# Nutrient and Rumen Fermentation Studies of Indian Pasture Legumes for Sustainable Animal Feed Utilisation in Semiarid Areas

**DOI:** 10.3390/ani13233676

**Published:** 2023-11-28

**Authors:** Sultan Singh, Tejveer Singh, Pushpendra Koli, Uchenna Y. Anele, Brijesh K. Bhadoria, Mukesh Choudhary, Yonglin Ren

**Affiliations:** 1ICAR-Indian Grassland and Fodder Research Institute, Jhansi 284003, India; singh.sultan@rediffmail.com (S.S.); tejveersinghbhu@gmail.com (T.S.); brijesh_bhadoria@yahoo.com (B.K.B.); 2College of Environmental and Life Sciences, Murdoch University, Murdoch, WA 6150, Australia; 3Department of Animal Sciences, North Carolina Agricultural and Technical State University, Greensboro, NC 27411, USA; uyanele@ncat.edu; 4ICAR-Indian Institute of Maize Research, Ludhiana 141001, India; mukeshagri08@gmail.com; 5School of Agriculture and Environment, The UWA Institute of Agriculture, The University of Western Australia, Perth, WA 6009, Australia

**Keywords:** animal feed, annual legumes, perennial legumes, nutritional value, minerals, in vitro fermentation, methane production

## Abstract

**Simple Summary:**

This research addresses the challenge of understanding the nutritional value of 16 Indian legume plants (5 annual and 11 perennial) for animal feed. The lack of comprehensive knowledge on their specific contributions and potential combinations for optimal animal nutrition poses a problem. The main objective is to assess the nutritive value, dry matter, mineral content and fermentation characteristics of these legumes. The study aims to fill the gaps in our understanding, exploring differences in protein, fibre and energy content among legumes. Additionally, it investigates gas and methane production to identify environmentally friendly options. The goal is to provide insights into how these legumes can be effectively combined, offering a scientific foundation for improving animal feeding practices in a sustainable and efficient manner.

**Abstract:**

This study evaluated 5 annual and 11 perennial Indian pasture legumes species for their nutritive value, dry matter and mineral contents and in vitro fermentation parameters. Legume species differed significantly (*p* < 0.05) in various nutritional aspects such as organic matter, crude protein (CP), ether extract, fibres and protein fractions. Perennial *Clitoria ternateaa* had higher (*p* < 0.05) buffer soluble protein (477), while neutral detergent soluble protein was highest in annually grown *Lablab purpureus* (420 g/kg CP). *Atylosia scarabaeoides* (AS) had higher levels of nonstructural carbohydrates (NSCs) (392 g/kg dry matter (DM)) than structural carbohydrates (SC) (367 g/kg DM). Its rapidly degradable fraction (51.7 g/kg (total carbohydrate) tCHO) was lower (*p* < 0.05) than other fractions of carbohydrates. Total digestible nutrients, digestible energy and metabolisable energy varied, with *Desmodium virgatus* (DV) having higher values and *Stylosanthas seabrana* (SSe) having the lowest. Predicted dry matter intake, digestible dry matter and relative feed value also showed significant differences (*p* < 0.05). Annual grasses such as *Dolichos biflorus*, *Macroptilium atropurpureum*, *Rhynchosia minima* (RM) were found to be better balanced with micro minerals. In vitro dry matter degradability, partition factor, short-chain fatty acids and microbial protein production of legumes varied significantly (*p* < 0.05). Gas and CH_4_ production (mL/g and mL/g (digestible DM) DDM) also varied, with *Clitoria ternatea*-blue having the highest gas production and *C. ternatea* -white (CT-w) and AS having lower CH_4_ production. Methane in total gas was low for DV, RM and CT-w (8.99%, 9.72% and 9.51%). Loss of DE and ME as CH_4_ varied (*p* < 0.05) among the legumes. Each legume offers unique benefits, potentially allowing for tailored combinations of annual and perennial legumes to optimize rumen feed efficiency.

## 1. Introduction

Inadequate feed and forage availability coupled with low nutritive value (crop residues, pasture grasses and grazing lands) are prime factors for poor animal production in the tropics and subtropics. For ages, forage legumes have played an important role in dairy and meat production [1]. Forage legumes species usually have high feeding value, as they are generally rich in protein and provide a substantial amount of energy, minerals and vitamins along with higher intake and digestibility [2,3]. Even in grain-based feed lot livestock production, forage legumes can make an economical contribution to animal health [4]. In recent years, many researchers have reviewed the role of forage legumes in supporting mixed crop–livestock production, the maintenance of pasture and grazing lands, and ecosystem services [5,6,7]. Dietary supplementation with forage legumes not only increases livestock production primarily via higher intake, nutrient content, and digestibility than cereal crops such as grasses, sorghum, maize, etc. [8,9], but also improves the rumen with fermentation efficiency via enhanced metabolisable energy, protein ratios and ruminal bypass protein availability to animals [10]; increased N retention [11]; and reduced methane emissions [8,12,13]. Combining annual forage legumes with crops such as maize may present an enhanced feeding option for animals in arid regions [14]. Further, the introduction of forage legumes in grass production areas as a grass–legume mixture could be one of the promising strategies to mitigate greenhouse gas (GHG) emissions from pasture-based livestock production [7]. The Food and Agriculture Organization of the United Nations (FAO; www.feedipedia.org accessed on 16 June 2018) lists around 169 legume species being used as forage and about 20 Mha land area are under forage legume monoculture [15]. Species of *Aeschynomene*, *Arachis*, *Centrosema*, *Desmodium*, *Macroptilium* and *Stylosanthes* offer promise for improved tropical pasture systems [16]. Alongside the wider morphological genetic variability in legume species [17], there exists nutritional variability [18]. In addition to genetic differences, the nutritional composition of legumes is also influenced by the season and growing location. It is important to obtain more information on legumes’ level of supplementation and nutritive value, as evaluation of their chemical composition has a substantial impact on the understanding of their nutritional value and also a great influence on animal nutrition [8,19]. Formulating specific diets for animal species at different physiological stages is influenced by the need to consider all parameters, including chemical composition (CP, NDF, ADF and lignin), total digestible nutrients (TDN) and in vitro digestibility [20,21]. 

The lack of comprehensive knowledge regarding the specific contributions and optimal combinations of legumes in animal nutrition constitutes a significant research gap, specifically in the Indian subcontinent context. To address this, a focused study on nutritional and rumen fermentation parameters is essential to provide insights into effective legume combinations, ensuring balanced and beneficial diets for livestock. To this end, the present study was planned in order to evaluate 5 annual and 11 perennial legume species from nine genera for cell wall constituents, carbohydrate fractions, protein fractions, energy contents, minerals, intake, digestibility and in vitro fermentation patterns (gas and methane production) for their prudent use in ruminant diets.

## 2. Materials and Methods

### 2.1. Experimental Sites 

The study was carried out at the Plant–Animal Relationship Division, ICAR-Indian Grassland and Fodder Research Institute, Jhansi, India. Laboratory procedures and animal management for donor sheep were carried out as per Institute animal ethics committee guidelines.

### 2.2. Sample Collection and Processing of Forage Legumes

Homogeneous plant samples (Table 1) were collected randomly from three different plots (30 × 10 m), maintained by “Grassland and Silvipasture Management Division of Institute”, for each legume. Samples of perennial legumes were collected after post-rainy season re-growth at flower initiation, while annual legumes samples were collected at the flower initiation growth stage. Plant samples were first air-dried in the shade on a cemented surface, followed by further drying in a hot-air oven at 60 °C for 72 h. Subsequently, the dried samples were ground using a 1 mm sieve in a Wiley mill.

### 2.3. Chemical Analyses

Samples for dry matter (DM), crude protein (CP), ether extract (EE) and ash were estimated as per the methods of the Association of Analytical Chemistry (AOAC) (1995). The CP of samples was estimated, per Kjeldahl, N × 6.25 via digestion in sulfuric acid and digestion mixture (consisting of sodium/potassium sulphate and copper sulphate in 10:1 ratio) using a semi-auto analyser (Kel Plus Classic-DX, Pelican, Chennai, India). The EE was determined by refluxing samples in petroleum ether using extraction apparatus. For ash estimation, samples were put in tarred silica basins and dried, and then basins were put into a muffle furnace at 600 °C for 4 h [22]. Neutral detergent fibre (NDF), acid detergent fibre (ADF), cellulose and lignin (ADL) were estimated via the sequential procedure modified by Van Soest et al. [23] a using fibre analyser (Fibra Plus FES 6, Pelican, Chennai, India). Heat-labile alpha amylase and sodium sulphite were not used in the NDF solution. For lignin (ADL) estimation, the sample left after ADF estimation was treated with 72% H_2_SO_4_ followed by ashing in a muffle furnace.

### 2.4. Carbohydrate and Protein Fractionation 

Carbohydrate fractions were estimated according to the Cornell Net Carbohydrate and Protein System (CNCPS) [24]. This is broadly classified into 4 fractions as follows:(a)C_A_: rapidly degradable carbohydrates (CHOs), including sugars.(b)C_B1_: intermediately degradable starch and pectin.(c)C_B2_: slowly degradable cell wall.(d)C_C_: unavailable/lignin-bound cell wall.

Total carbohydrates (tCHO g/kg DM) were determined by subtracting grams of CP, EE and ash contents from 1000. Structural carbohydrates (SCs) were calculated as the difference between NDF and neutral detergent insoluble protein and non-fibre carbohydrates were estimated as the difference between total CHOs and SCs [25]. Starch was estimated from samples after extraction with 80% ethyl alcohol to solubilize free sugars, lipids, pigments and waxes. The residue rich in starch was solubilized with perchloric acid and the extract was subjected to anthrone-sulphuric acid treatment for colorimetric determination of glucose, using a UV spectrophotometer (LABINDIA3000) at 630 nm [26]. 

Protein fractions of samples were partitioned into five fractions according to Licitra et al. [27]. These are as follows:

P_A_: non-protein nitrogen (NPN), the difference between total nitrogen and true CP nitrogen precipitated with sodium tungstate (0.30 M) and 0.5 M sulfuric acid.

P_B1_: buffer-soluble protein, the difference between true protein and buffer-insoluble protein, estimated with borate-phosphate buffer (pH 6.7 to 6.8) and freshly prepared 10% sodium azide solution. 

P_B2_: neutral detergent-soluble protein (NDSP), buffer-insoluble protein minus ND-insoluble protein.

P_B3_: acid detergent-soluble CP, the difference between ND-insoluble protein and acid detergent insoluble CP. 

P_C_: indigestible (this fraction contains protein associated with lignin, tannin–protein complexes and Maillard products that are unavailable to the animals. It is insoluble in the acid detergent).

### 2.5. Dry Matter Intake, Digestibility and Energy Calculations

Legumes’ dry matter intake (DMI), digestible DM (DDM), relative feed value (RFV), total digestible nutrients (TDN) and net energy for different animal functions, i.e., net energy of lactation (NEl), gain (NEg) and maintenance (Nem), were calculated using the following equations: (DMI = 120/NDF; DDM = 88.9 − 0.779 × ADF; RFV = (DDM × DMI) × 0.775; TDN = 104.97 − (1.302 × ADF); NEl = (TDN × 0.0245) − 0.012; NEg = TDN × 0.029) − 1.01; NEM = (TDN × 0.029) − 0.29) per Undersander et al. [28]. Digestible energy (DE, KJ/g DM; DE = TDN × 0.04409) and metabolizable energy (ME, KJ/g DM) values were calculated using the equations of Fonnesbeck et al. [29] and Khalil et al. [30], respectively. Metabolizable energy was calculated as 0.821 × DE.

### 2.6. Estimation of Minerals

For mineral estimation, legume samples were wet-digested with 3:1 HNO_3_: perchloric acid mixture, cooled and filtered through Whatman 42 filter paper. An aliquot was used for estimation of calcium (Ca), magnesium (Mg), copper (Cu), zinc (Zn), iron (Fe) and manganese (Mn), using an atomic absorption spectrophotometer (Varian AA 240) against the standards provided by Varian of sigma brand [31].

### 2.7. Donor Animals and Inoculum Preparation 

Four male adult *Jalauni* sheep, with mean body weight of 36.2 kg, were used as inoculum donors. These animals were maintained on a solely berseem hay diet and had free access to clean drinking water. Rumen liquor was collected in a pre-warmed thermos from each animal before feeding using a perforated tube from the stomach with the help of vacuum pressure pump. Collected rumen liquor was filtered through four layers of muslin cloth, and the strained rumen liquor obtained from different animals was mixed, kept at 39 °C in water bath and gassed with CO_2_ until used for mixing with incubating buffer media.

### 2.8. In Vitro Incubations

In vitro gas production was determined according to the pressure transducer technique of Theodorou et al. [32]. The incubation medium (1 L) was prepared via sequential mixing of buffer solution, macro-mineral solution, micro mineral solution and resazurin solution. Incubation medium was fluxed with CO_2_ till the pink colour turned colourless, and then 250 mL rumen liquor was added to obtain incubation medium with a rumen liquor ratio of 80:20. Samples (0.5 g) of air-dried legume samples were weighed into three 100 mL serum bottles. Three serum bottles without substrate/sample were used as blanks. Sample and control serum bottles were initially gassed briefly with CO_2_ before 50 mL of inoculum medium was added. Bottles were continuously fluxed with CO_2_ and then sealed with aluminium crimps. Before incubation, the gas pressure transducer was used to adjust head space gas pressure in each bottle to adjust the zero reading on the LED display and then incubated at 39 °C for 24 h to estimate the total gas production.

### 2.9. Methane Measurements

Methane in total gas for each of the legume samples, measured at 24 h of incubation from three bottles, was analysed via gas chromatography (Nucon 5765 Microprocessor controlled gas chromatograph, Okhla, New Delhi, India) equipped with a stainless-steel column packed with Porapak-Q and a flame ionization detector. One mL of gas, sampled using a Hamilton syringe from total gas produced, was injected manually (pull-and-push method of sample injection) into the gas chromatography (GC) equipment, which was calibrated with standard CH_4_ and CO_2_. Methane was also measured from blank bottles incubated for 24 h and used for correction of CH_4_ produced from the inoculum. Methane measured was related to total gas to estimate its concentration [33] and converted to energy and mass values using 39.54 kJ/L CH_4_ and 0.716 mg/mL CH_4_ factors, respectively [34]. Short-chain fatty acids (SCFA) were calculated using 24 h gas production [35], while partition factor (PF) and microbial mass (MBM) were estimated as described in previously described method [36].

### 2.10. Data Analysis

Each legume was replicated three times for every analysis group, and this consistent approach was maintained throughout the entire study. The means of nutritional and gas fermentation parameters of legumes were compared using one-way analysis of variance (ANOVA) with the statistical package for social sciences (SPPS) (version 16), with legumes as a fixed factor and parameters as dependent variables. Post hoc multiple comparison was performed using Duncan’s multiple range test to differentiate the means at *p* < 0.05 level. 

## 3. Results

### 3.1. Chemical Composition

Table 2 displays nutritional composition data for various legumes, highlighting OM, CP, EE, NDF, ADF, cellulose and lignin contents. The mean values across all legumes show an average organic matter content of 906, crude protein of 125.4, ether extract of 33.4, NDF of 549, ADF of 382, cellulose of 280.09, lignin of 93.9 and hemicellulose of 168 g/kg DM. Interestingly the protein contents were > 150 g/kg DM in *CP, CT-b* and *LLP* and lower < 100 g/kg DM in *DB, SSe, SH, SSc* and *SSco* (87.9, 80.4, 93.5, 92.2 and 94.8 g/kg DM). The EE contends were higher in *CT-w*, *CT-b* and *DV* (48.3, 46.6 and 63.8 g/kg DM), while lignin contents were lower in *LLP* (62.7) and *AH* (68.1), while the highest were in *SSe* (134) and *CT-w* (124 g/kg DM). *Stylosanthes* species had higher ADF (401 to 508) and cellulose (297 to 365 g/kg DM) contents, except (*DB*), than other evaluated legumes. 

### 3.2. Protein and Carbohydrate Fractions

Contents of tCHO, SC and NSC in legumes varied (*p* < 0.05), and their mean values were 747, 484 and 263 g/kg DM, respectively. Amongst the evaluated legumes, *AS* is the legume which had higher NSC than SC (392 vs. 367 g/kg DM Table 3). SC contents were highest in *DB* and lowest in AS (684 vs. 367 g/kg DM). Carbohydrate fractions of C_A_, C_B1_, C_B2_ and C_C_ differed (*p* < 0.05) in legumes and ranged between 143 and 589, 33.5 and 84.7, 4.30 and 551 and 221 and 428 g/kg tCHO, respectively. Rapidly degradable carbohydrate fraction in C_B1_ (51.7) was lower (*p* < 0.05) than in C_A_ (405), C_B2_ (242) and C_C_ (302 g/kg tCHO), respectively. 

Legume protein fractions P_A_, P_B1_, P_B2_, P_B3_ and P_C_ differed (*p* < 0.05 Table 3), and mean values were 226, 295, 209, 167 and 102 g/kg CP, respectively. The highest (295) accumulation of rapidly degradable protein fraction was found in P_B1_, compared to the lowest (1023 g/kg CP) of lignin-bound protein fraction (Pc). *Clitoria* species (CT-w and CT-b) and *MA* had higher P_B1_ fractions (437, 477 and 430) and the lowest P_C_ fractions (52.9, 71.6 and 75.2 g/kg CP).

### 3.3. Energy, Energy Efficiency, Intake, Digestibility and Relative Feed Value of Legumes

The TDN, DE and ME contents of legumes varied (*p* < 0.05) and their values were higher for *DV* (703, 12.9 and 10.6) and lowest for *SSe* (398, 7.28 and 5.99 kJ/g DM Table 4). Similarly, the NE values of legumes for lactation (NE_L_), maintenance (NE_M_) and gain (NE_G_) were higher for *DV* (6.66, 7.94 and 4.28) and lowest for *SSe* (3.53, 4.24 and 0.582 kJ/g DM), respectively. DMI, DDM and RFV of evaluated legumes differed (*p* < 0.05) with mean values of 2.22% body weight, 592 g/kg DM and 102%, respectively.

### 3.4. Minerals

Micro minerals (Cu, Zn, Fe and Mn) of legumes varied (*p* < 0.05) within ranges of 11.11–76.74, 20.77–65.85, 31.53–1288.04 and 18.61–70.51 ppm, respectively (Table 5). Ca and Mg varied from 0.55 to 2.77 and 0.25 to 0.88% with mean values of 1.26 and 0.42%, respectively.

### 3.5. Fermentation Pattern

Fermentation parameters (DMD, ME, PF, SCFA and MBP) of the legumes incubated in sheep inoculums varied (*p* < 0.05 Table 6), and their mean values were 578 g/kg DM, 5.88 kJ/g DM, 5.61 mL/mg DM, 2.42 mm/g and 351.65 mg/g, respectively. Microbial protein production efficiency varied (*p* < 0.05) from 0.32 for DV to 0.68 for *CT-w*.

### 3.6. Gas, Methane and Loss of Energy as Methane

In vitro gas and methane production (mL/g and mL/g DDM) of legumes varied (*p* < 0.05 Table 7). Gas production (mL/g) was lowest from *AS* (55.5) and highest from *CT-b* (141), while CH_4_ production was lowest (*p* < 0.05) from *CT-w* and *AS* (8.24 and 9.14) and highest from *AG* and *AH* (15.2 and 15.1). Methane in total gas varied (*p* < 0.05) and was low for *DV*, *RM* and *CT-w* (8.99, 9.72 and 9.51%) and high for *MA*, *AG* and *AH* (14.1, 14.0 and 13.9%), respectively. The loss of DE and ME as CH_4_ varied (*p* < 0.05) amongst the legumes, with mean values of 4.62 and 7.86%, respectively.

## 4. Discussion

### 4.1. Chemical Composition

Information on the chemical composition of forage has significant impact on the understanding of their nutritive value and animal production [8,19]. Legumes evaluated in present study except *DB* and *Stylosanthes* species had CP contents above the 110 g/kg DM recommended to fulfil the protein requirement of growing cattle. The crude protein and NDF content values for *Arachis*, *Centro*, *Stylo* and *Siratro* legumes ranged from 129 to 191 g/kg DM and 452 to 592 g/kg DM, respectively [37]. In our research, the values for *Stylosanthes* species and *AS* aligned with the findings of the aforementioned study. The earlier reported CP contents of *C. pubescens* (221) and *S. guyyanensis* (179 g/kg DM) were higher than our values (172 and 80.4–105 g/kg DM), respectively [18]. The CP, EE, NDF, AFD, cellulose and lignin contents of *C. pubesecens* were similar to earlier reports [18,38]. Further, the EE contents of *C. pubescens* and *S. guyanensis* (24.0 and 47.0 g/kg DM) of the above report were similar to the values of *CPb* (25.0) and *SSc* (43.2 g/kg DM), respectively. The levels of OM, NDF and ADF in *C. pubescens*, *S. hamata* and *S. scabra* were found to be consistent with the findings of the study conducted by Musco et al. [39]. However, the lignin contents of *C. pubescens* (167) and *S. scabra* (187) were higher than our values (103 and 72.9 g/kg DM), respectively. The NDF and ADF contents of *C. pubescens*, *Macroptelium bracteatum* and *M. gracile* was on par with a previous study [40]. Similarly, the range of CP, EE, OM, NDF, ADF, cellulose and lignin contents in *Macroptilium* species, *Rhynchosia minima*, *S. humilis*, and *Clitoriaternatea* were found to be in agreement with other studies [41,42]. In contrast, significant differences in the DM contents (154–253 g/kg DM) of 13 legumes were also recorded previously [43]. The CP content in 24 accessions of *Arachis* species ranged from 147 to 225.5 g/kg DM [44] which was like our results whereas in a study it was recorded even at higher range (184–250 g/kg DM) [45]. The OM and lignin contents (873–919 and 63–82 g/kg DM, respectively) closely resembled the values presented in the current study. The CP (73–129), EE (15–36) and ADF (373–424) except NDF (416–510 g/kg DM) of ten *Stylosanthes guyanensis* varieties recorded by Li et al. [46] were within the range of our observed values of *Stylosantehs* species. Furthermore, the chemical composition recorded in *Arachis hypogeal*, *C. pubescens*, *Clitoria ternatea*, *M. atropurpureus* and *S. guyanensis* were found to be similar to those of of earlier studies [47].

### 4.2. Protein and Carbohydrate Fractions

The protein composition of feeds reflects potential rumen degradation rates that estimate dietary nitrogen efficiency. The utilization of the nitrogenous fraction is important for evaluating feeds and specifying the nutritional requirements of ruminants [48]. Protein fractions (P_B1_, P_B2,_ P_B3_ and P_C_) differed (*p* < 0.05) across legumes and may be attributed to differences in concentrations of CP and lignin. Approximately 50–150 g/kg of the total nitrogen in forage is bound to lignin, making it unavailable to ruminal microorganisms, [49] and our legumes’ values for P_C_ lies within this range (52.9 to 150 g/kg CP). The P_C_ fraction of ten accessions of *Arachis pintoi* in the range of 178–276 g/kg CP [45] was higher than the values observed in our *Arachis species*.

Carbohydrates constitute the main energy source of plants (500–800 g/kg DM) and play an important role in animal nutrition as a prime source of energy for rumen microorganisms [49]. Carbohydrate accumulation in fodder crops is influenced by several factors, such as plant species, variety, growth stage and environmental conditions during growth [50]. Mlay et al. [51] reported a mean tCHO content of 746 g/kg DM for *Macropttlium atropurpurus*, which was higher than our values (689 g/kg DM). The higher tCHO contents for blue (800) and white (830 g/kg DM) varieties of *Clitoria ternatea* than our *CT-w* (725) and *CT-b* (708 g/kg DM) values was reported earlier [41]. The tCHO (626–701 g/kg DM) for ten *Arachis pinto* accessions were found to be lower than our *AG* and *AH* values [45]. Higher SC in DB, SSe, SH and RM may be due to the higher NDF contents recorded in the present study. The legumes *AG*, *AH*, *CPb*, *MA*, *CT-b* and *AS* had higher values of C_A_ fraction (> 44 g/kg tCHO), and feeds higher in this fraction are considered good energy sources to stimulate rumen microorganism growth [52] and the synchronism between protein and carbohydrate digestion rates, having an important effect on the end products of fermentation and on animal production [53]. Low contents of unavailable carbohydrate fraction (C_C_) in *AH*, *DB*, *SSe*, *DV*, *SV* and *LLP* may be due to their lower lignin contents. This indigestible fraction with C_B2_ usually affects animal intake by the rumen fill, which can reduce animal performance [54]. In our study we recorded that *DB*, *DV*, *RM*, *SH* and *LLP* legumes with higher hemicelluloses contents (315, 253, 258, 221 and 220 g/kg DM) had higher C_B2_ fraction contents (551, 296, 254, 365 and 388 g/kg tCHO), which are more slowly degraded in the rumen, impacting microbial synthesis and animal performance [55]. Higher hemicellulose concentrations result in higher concentrations of C_B2_ fraction. Carvalho et al. [52] reported that NDF concentration influences carbohydrate fraction C_B2_, and forage high in NDF concentration usually has higher values of C_B2_. Our results partially agree with these observations that most of the legumes with higher NDF contents had higher C_B2_ fraction value. Moreover, the annual forage RM, abundant in flavonoids, has proven to be suitable for long-term enhancement of rangelands [56,57]. The observed variations are likely attributed to differences in sampling time, growth stage and climatic conditions.

### 4.3. Energy, Energy Efficiency, Intake, Digestibility and Relative Feed Value of Legumes

Feed or fodder nutritive value is a function of its dry matter intake and its ability to provide the nutrients in the right proportion required by animals for different physiological functions [58]. Greater calculated DMI value for *AG*, *AH*, *MA* and *DV* (2.51, 2.51, 2.50 and 2.73%) than other legumes (1.64–2.31%) may be due to lower NDF contents. The NDF of forage has been negatively correlated with DMI, which is not always consistent, although NDF is positively related with resistance to ruminal degradation [59]. The lowest and highest DMD of *SSe* and *DV* legumes may be due to their highest (501) and lowest (266 g/kg DM) ADF contents, respectively. Therefore, careful consideration of these factors is essential in optimizing the nutritional value of feed or fodder for animal consumption.

Musco et al. [39] reported higher NE_L_ values for *C. pubescens*, *S. hamata* and *S. scabra* (9.53, 6.16 and 5.49 kJ/g DM) than our recorded values (5.16, 4.88 and 4.58 kJ/g DM), respectively. Similarly, ME values of *C. pubescens* were higher (6.53) than our values (5.78 kJ/g DM) [60]. The ME contents (9.44–10.36 kJ/g DM) of 13 legumes were higher than ME values observed in the present study [43]. TDN consists of digestible nutrients that are available for livestock and is primarily related to forage ADF contents. As ADF increases, there is a corresponding decrease in TDN, leading to the unavailability of forage nutrients to animals [61]. Relatively higher values for TDN (634 g/kg DM), GE (17.2), DE (11.5) and ME (9.44 kJ/gDM) were recorded than the values of *MA* in present study [51]. 

The *IVDMD* of *C. pubescens* and *S. guyanensis* (530 and 570 g/kg DM) reported earlier was similar to our values of *CPb* (513) and *SSe* and *SV* (578 and 580 g/kg DM) but lower than *SH* (641), *SSc* (631) and *SSco* (646 g/kg DM), respectively [18], reported by us. The OMD of *C. pubescens* (478), *S. hamata* (609) and *S. scabra* (593) were relatively higher than our values (457, 447 and 471 g/kg DM, respectively). Tona et al. [38] reported higher ME and OMD for *C. pubescens* than our values. The DMD and OMD (614–712 and 586–688 g/kg DM) for five forage legumes [40] were more than the range values (395–663 and 375–526 g/kg DM) of legumes evaluated in the present study. Similarly, the mean OMD and DMD (656 and 684 g/kg DM) were higher than the average OMD and DMD values of tested legumes. The higher values for OMD (642–739 g/kg DM) in legume forage than our values mentioned earlier [43]. Fernandes et al. [62] recorded DMD of ten *Arachis* species over three years in range of 501–632 g/kg DM. The values of IVOMD between 600 to 740 g/kg DM for 24 accessions of *Arachis* pinto substantiate our DMD values of *Arachis* species [44]. The variations in the present study’s highest and lowest total digestible nutrient (TDN) values for *DV* and *SSe* could be linked to differences in their digestibility. Specifically, the higher ADF content in *SSe* and the lower ADF content in *DV* may be influencing these TDN values, as an increase in ADF tends to lead to a decrease in TDN values.

NDF and ADF contents are negatively associated with the OMD and ME values of legumes [43]. Intake, an important part of forage nutritive value, is partially related to cell wall content and bulkiness of forage. The lowest value of DMI for *DB* (1.64) versus the highest of AS (2.73%) may be attributable to their maximum (730) and minimum NDF contents (440 g/kg DM) as the NDF contents are negatively associated with DMI [49].

The relative feed value (RFV) of ten varieties of *S. guyanensis* in a range of 102.8–130.5% [46] were higher than the values determined for *Stylosanthes* species (70.80–101.46%). The values of *IVDMD* and TDN of *A. hypogeaea* (78.86 and 71.44), *C. pubescens* (59.89 and 51.30), *Clitoria ternatea* (74.15 and 60.20) *M. atropurpureus* (66.51 and 56.15) and *S. guyanensis* (66.22% and 54.43%) are higher than our values for these legumes [63]. The DMD of *Arachis* species (*AG* and *AH*) was higher (650 and 663 g/kg DM) except *LLP* than other legume species, which is similar to previous observations [64] where it was reported that the DMD of *Arachis* species in general is greater than other tropical legumes. 

### 4.4. Minerals 

The Ca contents (0.50–0.80%) of legumes studied here were found to be relatively lower than reported earlier [18], while Mg contents (0.32–0.63%) were like our values of legumes. Similar trends of Ca and Mg contents were reported in at least one other study [65]. Legumes’ Fe, Cu, Mn and Zn contents in the ranges of 441–494, 2.42–7.14, 55.2–61.3 and 34.8–65.3 ppm [65] were more or less similar to our micro mineral values. In contrast, relatively lower Ca (1.01–1.62%) and higher Mg (0.92–1.96%) in different *Arachis* pinto cultivars in humid and sub-humid environments were reported than our values [66]. Further, the Cu, Fe, Zn and Mn reported for this legume were inconsistent with our values. Macro (Ca and Mg) and micro minerals (Cu. Zn, Fe and Mn) of the white and blue varieties of *Clitoria ternatea* reported earlier [41] were more or less similar to our CT-w and CT-b values. Fernandes et al. [62] reported Ca and Mg values in ranges of 1.4–2.47% and 0.08–0.59% for ten *Arachis* species over three years, similar to our *Arachis* species values. The macro (Ca and Mg) and micro minerals (Cu, Fe, Zn and Mn) of four herbaceous legumes and two browse legumes reported by [66] were in partial agreement with our mineral values. The variation in mineral contents of legume species may be due to differences in soil mineral contents, growth stage, fertilizer application and environmental conditions [67].

### 4.5. Fermentation Pattern

The 1.15 µmole SCFA content of *C. pubescens* at 24 h of fermentation was reported on earlier [38]. The partition factor in the range of 3.07–4.94 mg/mL reported earlier [39] was lower than the PF values (3.45–7.08) of legumes evaluated in present study. The reason for higher PF values for evaluated legumes may be ascribed to lower gas production due to lower DMD. 

Partition factor, an indicator of the fermentation efficiency of a feedstuff, is expressed as volume of gas (mL) produced per mg of substrate degraded. Legumes’ mean PF values were higher (5.61) than the theoretically possible maximum value (4.14) [36]. This suggests that the unique nutritional composition, potentially lower fibre content, and favourable substrate characteristics in legumes contribute to enhanced microbial fermentation efficiency in the rumen, underscoring the importance of these factors in understanding and optimizing feedstuff utilization in ruminant nutrition. The higher PF values for CT-w (7.05) could not translate to higher microbial mass, probably due to their lower gas production. Higher SCFA for DV, LLP, SSc and RM may be due to their higher gas production as reported. For *DV*, microbial protein production was lowest (143 mg/g) and methane production was highest (26.45 mL/g DDM), which is consistent with previous observations [68]. Greater PF values for CT-w did not translate to greater microbial mass as noted in a previous study [69]. We expected higher PF to translate to greater microbial mass, as PF is the measure of efficiency of microbial production. We can only speculate the reason for this observation, but the higher SCFA value noted for *DV* is consistent with a previous report that microbial mass and SCFA are inversely related [70]. It may be that DV partitioned more energy into SCFA versus microbial mass production, while in the case of LLP, energy partition was well distributed between SCFA and microbial mass production.

### 4.6. Gas, Methane Production and Loss of Energy as Methane

One of the options to improve feed efficiency via efficient rumen fermentation is that dry matter conversion to methane is less and utilization by animal is more. Methane is an end-product of rumen fermentation and causes loss of dietary energy [71]. This loss of feed energy as methane varies with its quality [72,73,74,75] and with animal species. Methane % of gas between 15.9 to 18.4%) from 13 legumes [43] was higher than our recorded values for 16 legumes (9.0 to 14.1%). Our values for proportion of methane in total gas except for DV, RM and CT-b were within the values (12.3–15.9%) reported earlier [65]. Lopez et al. [76] categorised methane reduction potential as low (% of CH_4_ in gas between 11 and 14%). Thus, the legumes evaluated in the present study had low methane production. A higher gas production for *S. hamata* and *S. scabra* (195 and 193) than *C. pubescens* (136 mL/g OM) were reported, and these values were higher, with a similar gas production pattern [39]. Similarly, a report on higher gas production at 24 h for *C. pubescens* (25.4 mL/200 mg) than our values was published [60]. Like our observations, Suha et al. [43] recorded significant difference in gas production (42.56 to 51.42 mL/200 mg) from the hays of 13 legumes, and these values were higher than our values (55.46 to 141.26 mL/g) from 16 legumes, and these differences may be due to our lower OMD values as low OM fermented for gas production. Further the methane production in range of 7.36–8.78 mL/200 mg was higher than our methane production values. Variations in the gas and methane production among the assessed pasture legume species can be attributed to differences in their chemical composition and degradation rates. Such variation in methane and gas production of legume species has been recorded earlier [43,77,78,79]. 

The proportion of methane to total gas production is an important indicator of methane emission potential of feed/fodder degradation. This ratio of methane to gas for common feeds (hays, concentrate, mixture of hays and concentrate) vary between 16–20%. Methane production from *Desmodium intortum* (3.67), *Medicago sativa* (5.90) and *Vicia sativa* (5.73 mL/200 mg DM) legumes observed by [65] was also higher than that of most of our legumes but similar to that of *Arachis* species. Maccarana et al. [80] reported gas production, methane production and CH_4_% of total gas in range of 72–480 mL/g DM, 7.3–77.5 mL/g and 9.4–40.61% from 390 observations of 30 studies. Relatively lower values of gas production, methane production and CH_4_% total gas in the present study may be due to sheep rumen liquor, as values of gas and methane production and CH_4_% total gas were higher when bovine rumen liquor was used for incubations [81,82]. Greater methane production may be attributed to their higher *IVDMD* values. In line with present findings, previous studies have reported that feedstuffs with higher gas production and *IVDMD* tended to have higher methane production per gram DM incubated [83]. Methane emission differences within legumes may be attributed to the variability in their chemical constituents as reported earlier [53,84,85]. The differences in relative proportion of CH_4_ to energy values (CH_4_% of DE and % of ME) may be attributed to variations in legume energy values, level of methane production and dry matter degradability. Variability in proportion of methane to energy of feeds and fodders has been reported earlier [53,65].

## 5. Conclusions

This study fills a large gap in our understanding of the potential combination of sub-tropical legumes for grazing animals. Notably, AG, AH, DV and AS contained high energy along with lower structural carbohydrates, making them promising as potential livestock feeds. Additionally, DV and RM produced low methane in vitro, suggesting potential environmental benefits. These findings enhance current nutrient knowledge of these pasture legumes and pave the way for a more sustainable livestock industry on the Indian sub-continent in the future. However, future studies on animal responses to high dietary inclusion levels are required.

## Figures and Tables

**Table 1 animals-13-03676-t001:** Legume species used in the study.

Annual species	*Dolichos biflorus* (DB), horse gram; *Lablab purpures* (LLP), Lablab bean; *Macroptilium atropurpureum* (MA) purple bush bean; *Rhynchosia minima* (RM), burn-mouth vine; and *Stylosanthas hamata* (SH), Caribbean stylo.
Perennial species	*Arachis glabrata* (AG), perennial peanut; *Arachis hagenbackii* (AH); *Atylosia scarabaeoides* (AS), showy pigeonpea; *Clitoria ternatea*-white (CT-W), white butterfly pea vine, *Clitoria ternatea*-blue (CT-B), butterfly pea; *Centrosoma pubescene* (CPb), fodder pea; *Desmenthus virgatus* (DV), hedge lucerne; *Stylosanthas scabra* (SSc), shrubby stylo; *Stylosanthas scofield* (SSco), Brazilian lucerne; *Stylosanthas seabrana* (SSe), cattinga stylo; and *Stylosanthus viscosa* (SV), sticky stylo.

**Table 2 animals-13-03676-t002:** Chemical composition of range legumes (g/kg DM).

Legumes	OM	CP	EE	NDF	ADF	Cellulose	Lignin	Hemi Cellulose
AG	873 ^b^	123 ^f^	20.2 ^ab^	479 ^b^	381 ^cd^	2725 ^e^	101 ^ef^	976 ^a^
AH	878 ^c^	116 ^e^	23.4 ^abc^	478 ^b^	342 ^b^	273 ^e^	68.1 ^ab^	136 ^bcd^
CP	905 ^g^	172 ^i^	27.5 ^cd^	549 ^cde^	380 ^c^	266 ^c^	103 ^ef^	169 ^d^
DB	897 ^f^	88 ^b^	28.1 ^cd^	730 ^h^	414 ^ef^	325 ^e^	82.4 ^bcd^	316 ^g^
CT-w	919 ^i^	146 ^h^	48.3 ^f^	566 ^f^	396 ^cde^	270 ^c^	124 ^gh^	171 ^d^
SSe	930 ^L^	80 ^a^	26.6 ^cd^	604 ^g^	501 ^g^	365 ^g^	134 ^i^	103 ^ab^
SH	923 ^j^	93 ^c^	27.0 ^cd^	624 ^g^	403 ^e^	297 ^d^	99.2 ^e^	221 ^e^
SSc	915 ^h^	92 ^c^	43.2 ^ef^	548 ^cde^	423 ^f^	345 ^f^	72.9 ^abc^	124 ^abc^
SSco	886 ^d^	95 ^c^	18.5 ^a^	529 ^cd^	401 ^de^	307 ^de^	83.8 ^bc^	129 ^abc^
SV	927 ^k^	105 ^d^	24.6 ^bcd^	545 ^cde^	407 ^ef^	320 ^g^	80.7 ^bc^	139 ^bcd^
MA	855 ^a^	141 ^g^	24.9 ^bcd^	482 ^b^	382 ^cd^	271 ^e^	105 ^ef^	99.0 ^a^
AS	894 ^e^	105 ^d^	29.7 ^d^	439 ^a^	342 ^b^	213 ^b^	115 ^fg^	98.2 ^a^
DV	928 ^kl^	143 ^gh^	63.8 ^g^	519 ^c^	266 ^a^	177 ^a^	73.5 ^abc^	253 ^ef^
RM	934 ^m^	143 ^gh^	41.5 ^e^	595 ^fg^	337 ^b^	229 ^b^	1027 ^ef^	258 ^f^
CT-b	937 ^n^	183 ^j^	46.6 ^f^	544 ^cde^	396 ^cde^	296 ^d^	94.6 ^de^	147 ^cd^
LLP	896 ^e^	180 ^j^	40.3 ^e^	556 ^de^	335 ^b^	267 ^c^	62.7 ^a^	220 ^e^
Mean	906	125.4	33.4	549	382	28.09	93.9	168
SEM	2.12	0.283	0.421	1.002	1.51	6.93	1.69	2.87
Significance	<0.0001	<0.0001	<0.0001	<0.0001	<0.0001	<0.0001	<0.0001	<0.0001

AG, *Arachis glabrata*; AH, *Arachis hagenbackii*; CPb, *Centrosoma pubescene*; DB, *Dolichos biflorus*; CT-w, *Clitoria ternatea* (white); SSe, *Stylosanthas seabrana*; SH, *Stylosanthas hamata*; SSc, *Stylosanthas scabra*; SSco, *Stylosanthas scofield*; SV, *Stylosanthus viscosa*; MA, *Macroptilium atropurpureum*; AS, *Atylosia scarabaeoides*; DV, *Desmenths virgatus*; RM, *Rhynchosia minima*; CT-b; *Clitoria ternatea* (blue); LLP, *Lablab purpures*; OM, organic matter; DM, dry matter; CP, crude protein; EE, ether extract; NDF, neutral detergent fibre expressed inclusive residual ash; ADF, acid detergent fibre expressed inclusive of residual ash; Lignin(sa), lignin solubilized with sulphuric acid; SEM, standard error of means; superscripted letters indicate significant differences in values.

**Table 3 animals-13-03676-t003:** Carbohydrates (g/kg DM), their fractions (g/kg tCHO) and protein fractions (g/kg CP) of range of legumes.

Legumes	Carbohydrate and Its Fractions							Protein Fractions				
	tCHO	SC	NSC	C_A_	C_B1_	C_B2_	C_C_	P_A_	P_B1_	P_B2_	P_B3_	P_C_
AG	730 ^e^	424 ^bc^	306 ^ghi^	470 ^f^	71 ^g^	127 ^bc^	315 ^gh^	223 ^de^	394 ^fg^	161 ^cd^	116 ^cd^	98.8 ^d^
AH	739 ^f^	430 ^bcd^	309 ^ghi^	439 ^ef^	57 ^e^	282 ^gh^	221 ^a^	228 ^de^	374 ^ef^	224 ^e^	92.8 ^bc^	81.5 ^c^
CP	705 ^c^	457 ^cde^	248 ^def^	459 ^ef^	48 ^cd^	153 ^cd^	349 ^def^	214 ^cd^	185 ^b^	305 ^g^	202 ^ef^	93.7 ^d^
DB	781 ^i^	683 ^h^	976 ^a^	143 ^a^	52 ^d^	551 ^j^	253 ^a^	324 ^h^	123 ^a^	300 ^g^	118 ^cd^	134 ^fg^
CT-w	725 ^de^	500 ^f^	225 ^cd^	429 ^ef^	37 ^a^	121 ^bc^	428 ^g^	211 ^c^	437 ^h^	91.5 ^b^	208 f	52.9 ^a^
SSe	823 ^L^	561 ^g^	261 ^ef^	358 ^c^	43 ^b^	208 ^def^	391 ^fg^	184 ^b^	352 ^e^	172 ^d^	178 ^e^	113 ^e^
SH	802 ^k^	589 ^g^	213 ^bc^	278 ^b^	61 ^e^	365 ^i^	297 ^bc^	277 ^g^	232 ^c^	346 ^h^	60.8 ^a^	83.7 ^c^
SSc	780 ^j^	483 ^ef^	296 ^gh^	406 ^de^	37 ^a^	332 ^hi^	224 ^a^	236 ^ef^	182 ^b^	292 ^g^	139 ^d^	150 ^h^
SSco	773 ^i^	454 ^cde^	319 ^hi^	431 ^ef^	58 ^e^	251 ^efg^	260 ^ab^	238 ^ef^	194 ^b^	179 ^d^	246 ^g^	142 ^gh^
SV	797 ^k^	464 ^de^	333 ^i^	414 ^de^	65 ^f^	277 ^fgh^	243 ^a^	214 ^cd^	265 ^d^	138 ^c^	254 ^g^	129 ^f^
MA	689 ^b^	415 ^b^	274 ^fg^	478 ^f^	85 ^h^	70 ^b^	367 ^ef^	165 ^a^	430 ^h^	252 ^f^	78.3 ^ab^	75.2 ^cd^
AS	759 ^h^	367 ^a^	392 ^i^	589 ^g^	43 ^b^	4.3 ^a^	364 ^def^	250 ^f^	205 ^bc^	99.1 ^b^	330 ^h^	116 ^e^
DV	721 ^d^	458 ^cde^	263 ^ef^	410 ^de^	49 ^cd^	296 ^gh^	245 ^a^	215 ^cd^	281 ^d^	310 ^g^	110 ^cd^	82.9 ^c^
RM	749 ^g^	516 ^f^	233 ^cd^	375 ^cd^	45 ^bc^	254 ^efg^	326 ^cde^	210 ^c^	418 ^gh^	25.2 ^a^	245 ^g^	1015 ^d^
CT-b	708 ^c^	453 ^cde^	254 ^def^	450 ^ef^	42 ^b^	187 ^cde^	321 ^cd^	205 ^c^	477 ^i^	33.8 ^a^	213 ^f^	71.6 ^b^
LLP	676 ^a^	484 ^ef^	191 ^b^	356 ^b^	33 ^a^	388 ^i^	223 ^a^	215 ^cd^	177 ^b^	420 ^i^	76.8 ^ab^	111 ^e^
Mean	747	484	263	405	52	242	302	226	295	209	167	102
SEM	0.343	2.74	2.77	14.03	1.96	19.5	9.50	5.30	16.4	16.4	11.3	3.90
Significance	<0.0001	<0.0001	<0.0001	<0.0001	<0.0001	<0.0001	<0.0001	<0.0001	<0.0001	<0.0001	<0.0001	<0.0001

AG, *Arachis glabrata*; AH, *Arachis hagenbackii*; CPb, *Centrosoma pubescene*; DB, *Dolichos biflorus*; CT-w, *Clitoria ternatea* (white); SSe, *Stylosanthas seabrana*; SH, *Stylosanthas hamata*; SSc, *Stylosanthas scabra*; SSco, *Stylosanthas scofield*; SV, *Stylosanthus viscosa*; MA, *Macroptilium atropurpureum*; AS, *Atylosia scarabaeoides*; DV, *Desmenths virgatus*; RM, *Rhynchosia minima*; CT-b; *Clitoria ternatea* (blue); LLP, Lablab purpures; tCHO, total carbohydrates; SC, structural carbohydrates; NSC, non-structural carbohydrates; C_A_, rapidly degradable CHO including sugars; C_B1_, intermediately degradable starch and pectin; C_B2_, slowly degradable cell wall; C_C_, unavailable/lignin-bound cell wall; P_A_, non-protein nitrogen; P_B1_, buffer soluble protein; P_B2_, neutral detergent soluble protein; P_B3_, acid detergent soluble protein; P_C_, indigestible protein; SEM, standard error of means; superscripted letters indicate significant differences in values.

**Table 4 animals-13-03676-t004:** Energy value of range of legumes.

Legumes	TDN	DE	ME	NE_L_	NE_M_	NE_G_	DMI	DDM	RFV
AG	553 ^de^	10.1 ^e^	8.32 ^d^	5.16 ^f^	6.16 ^e^	2.45 ^de^	2.51 ^g^	592 ^de^	114.94 ^ef^
AH	604 ^f^	11.1 ^f^	9.11 ^e^	5.66 ^g^	6.74 ^f^	3.08 ^f^	2.51 ^g^	623 ^f^	121.26 ^fg^
CP	555 ^e^	10.2 ^e^	8.36 ^d^	5.16 ^ef^	6.16 ^e^	2.50 ^e^	2.19 ^def^	593 ^e^	100.45 ^cd^
DB	511 ^ab^	9.36 ^bc^	7.69 ^bc^	4.70 ^be^	5.62 ^bc^	1.96 ^bc^	1.64 ^a^	567 ^bc^	72.20 ^a^
CT-w	534 ^cde^	9.82 ^cde^	8.03 ^cd^	4.95 ^cdef^	5.91 ^cde^	2.25 ^cde^	2.12 ^cd^	581 ^cde^	95.48 ^c^
SSe	398 ^a^	7.28 ^a^	5.99 ^a^	3.54 ^a^	4.24 ^a^	0.58 ^a^	1.99 ^bc^	499 ^a^	76.80 ^a^
SH	525 ^c^	9.65 ^cd^	7.90 ^c^	4.87 ^cde^	5.78 ^c^	2.12 ^c^	1.93 ^b^	575 ^c^	86.16 ^b^
SSc	498 ^b^	9.150 ^b^	7.48 ^b^	4.58 ^b^	5.49 ^b^	1.83 ^b^	2.20 ^def^	559 ^b^	95.65 ^c^
SSco	528 ^cd^	9.69 ^cde^	7.94 ^cd^	4.87 ^cde^	5.82 ^cd^	2.16 ^cd^	2.27 ^ef^	577 ^cd^	101.46 ^cd^
SV	520 ^ab^	9.52 ^bc^	7.82 ^bc^	4.78 ^f^	5.74 ^c^	2.08 ^bc^	2.20 ^def^	572 ^bc^	97.68 ^c^
MA	552 ^de^	10.1 ^de^	8.32 ^d^	5.12 ^bc^	6.12 ^de^	2.45 ^de^	2.50 ^g^	591 ^de^	114.36 ^e^
AS	605 ^f^	11.1 ^f^	9.11 ^e^	5.66 ^g^	6.78 ^f^	3.08 ^f^	2.73 ^h^	623 ^f^	131.73 ^h^
DV	703 ^g^	12.9 ^g^	10.6 ^f^	6.66 ^h^	7.95 ^g^	4.28 ^g^	2.31 ^f^	682 ^g^	122.22 ^g^
RM	611 ^f^	11.2 ^f^	9.19 ^e^	5.74 ^e^	6.82 ^f^	3.16 ^f^	2.02 ^be^	627 ^f^	97.92 ^c^
CT-b	534 ^cde^	9.77 ^cde^	8.03 ^cd^	4.95 ^b^	5.91 ^cde^	2.25 ^cde^	2.21 ^def^	580 ^cde^	99.45 ^cd^
LLP	613 ^f^	11.2 ^f^	9.23 ^e^	5.74 ^e^	6.86 ^f^	3.20 ^f^	2.16 ^de^	627 ^f^	104.94 ^d^
Mean	553	10.15	8.40	5.12	6.12	2.47	2.22	592	102.04
SEM	2.04	0.029	0.029	0.021	0.025	0.025	0.011	1.22	0.548
Significance	<0.0001	<0.0001	<0.0001	<0.0001	<0.0001	<0.0001	<0.0001	<0.0001	<0.0001

AG, *Arachis glabrata*; AH, *Arachis hagenbackii*; CPb, *Centrosoma pubescene*; DB, *Dolichos biflorus*; CT-w, *Clitoria ternatea* (white); SSe, *Stylosanthas seabrana*; SH, *Stylosanthas hamata*; SSc, *Stylosanthas scabra*; SSco, *Stylosanthas scofield*; SV, *Stylosanthus viscosa*; MA, *Macroptilium atropurpureum*; AS, *Atylosia scarabaeoides*; DV, *Desmenths virgatus*; RM, *Rhynchosia minima*; CT-b; *Clitoria ternatea* (blue); LLP, *Lablab purpures*; TDN, total digestible nutrients; DE, digestible energy; ME, metabolizable energy; NE_L_, net energy for lactation; NE_M_, net energy for maintenance; NE_G_, net energy for growth; DMI, dry matter intake; DDM, digestible dry matter; RFV, relative feed value; SEM, standard error of means; superscripted letters indicate significant differences in values.

**Table 5 animals-13-03676-t005:** Macro (%) and micro minerals (ppm) of range of legumes.

Legumes	Cu	Zn	Fe	Mn	Ca	Mg
AG	76.7 ^h^	65.8 ^g^	1150 ^cde^	47.5 ^c^	2.77 ^f^	0.59 ^j^
AH	60.7 ^g^	61.1 ^g^	737 ^bc^	47.3 ^c^	2.74 ^f^	0.58 ^j^
CP	28.7 ^ef^	42.2 ^e^	246 ^ab^	31.9 ^ab^	1.48 ^e^	0.37 ^e^
DB	35.7 ^f^	31.9 ^bc^	1271 ^de^	70.5 ^ef^	1.37 ^e^	0.41 ^g^
CT-w	24.9 ^cde^	41.7 ^de^	311 ^ab^	40.8 ^bc^	0.60 ^a^	0.37 ^f^
SSe	14.9 ^ab^	35.0	215 ^ab^	20.3 ^a^	0.94 ^c^	0.24 ^e^
SH	11.1 ^a^	26.7 ^ab^	259 ^ab^	77.37 ^f^	0.80 ^b^	0.25 ^ab^
SSc	13.6 ^ab^	32.0 ^bc^	174 ^a^	22.1 ^a^	1.34 ^e^	0.28 ^bc^
SSco	16.1 ^abc^	48.7 ^f^	1023 ^cd^	54.5 ^cd^	1.47 ^e^	0.29 ^cd^
SV	12.4 ^ab^	20.8 ^a^	137 ^a^	27.2 ^a^	1.44 ^e^	0.46 ^h^
MA	21.1 ^bcde^	30.1 ^bc^	1573 ^e^	61.4 ^e^	1.17 ^d^	0.34 ^ef^
AS	18.5 ^abcd^	28.5 ^bc^	1288 ^de^	72.3 ^f^	0.93 ^c^	0.36 ^ef^
DV	27.3 ^def^	27.1 ^ab^	32.8 ^a^	28.3 ^ab^	0.93 ^c^	0.88 ^k^
RM	59. 6 ^g^	25.5 ^ab^	183 ^a^	18.6 ^a^	0.58 ^a^	0.32 ^de^
CT-b	72.0 ^h^	35.3 ^cd^	31.5 ^a^	41.9 ^bc^	0.55 ^a^	0.53 i
LLP	62.8 ^g^	39.3 ^de^	39.4 ^a^	41.6 ^bc^	0.98 ^c^	0.44 ^gh^
Mean	34.8	37.0	542	44.0	1.26	0.42
SEM	0.726	0.544	41.41	1.08	0.11	0.003
Significance	<0.0001	<0.0001	<0.0001	<0.0001	<0.0001	<0.0001

AG, *Arachis glabrata*; AH, *Arachis hagenbackii*; CPb, *Centrosoma pubescene*; DB, *Dolichos biflorus*; CT-w, *Clitoria ternatea* (white); SSe, *Stylosanthas seabrana*; SH, *Stylosanthas hamata*; SSc, *Stylosanthas scabra*; SSco, *Stylosanthas scofield*; SV, *Stylosanthus viscosa*; MA, *Macroptilium atropurpureum*; AS, *Atylosia scarabaeoides*; DV, *Desmenths virgatus*; RM, *Rhynchosia minima*; CT-b; *Clitoria ternatea* (blue); LLP, *Lablab purpures*; SEM, standard error of means; superscripted letters indicate significant differences in values.

**Table 6 animals-13-03676-t006:** Partition factor (PF), short-chain fatty acids (SCFA), degradable dry matter and microbial protein production from fermentation of range of legumes in sheep inoculum.

Legumes	ME kJ/g	DMD	PF	SCFA mm/g	MBM mg/g	EMBP mg/g
AG	6.07 ^cd^	649 ^fg^	6.09 ^cde^	2.44 ^cd^	421 ^efg^	0.63 ^cd^
AH	6.04 ^cd^	663 ^g^	6.35 ^cde^	2.41 ^cd^	445 ^g^	0.64 ^cd^
CP	5.77 ^bcd^	513 ^b^	5.35 ^bc^	2.20 ^bcd^	306 ^bc^	0.60 ^bcd^
DB	6.11 ^cd^	577 ^cd^	5.26 ^bc^	2.54 ^cd^	346 ^cde^	0.56 ^bcd^
CT-w	4.86 ^ab^	559 ^cd^	7.05 ^de^	1.80 ^ab^	392 ^defg^	0.68 ^d^
SSe	5.61 ^bc^	578 ^cd^	5.49 ^bc^	2.41 ^cd^	355 ^cdef^	0.59 ^bcd^
SH	5.82 ^bcd^	641 ^fg^	6.20 ^cde^	2.37 ^cd^	417 ^defg^	0.63 ^cd^
SSc	6.18 ^cd^	631 ^efg^	5.60 ^bcde^	2.61 ^de^	385 ^defg^	0.59 ^bcd^
SSco	5.78 ^bcd^	646 ^fg^	6.25 ^cde^	2.42 ^cd^	424 ^efg^	0.63 ^cd^
SV	5.84 ^bcd^	580 ^cd^	5.24 ^bc^	2.54 ^cd^	342 ^cd^	0.57 ^bcd^
MA	5.59 ^bc^	594 ^cde^	7.08 ^e^	2.00 ^abc^	407 ^defg^	0.66 ^d^
AS	4.49 ^a^	394 ^a^	5.53 ^bc^	1.64 ^a^	238 ^b^	0.58 ^bcd^
DV	6.53 ^cd^	409 ^a^	3.45 ^a^	2.77 ^de^	142 ^a^	0.32 ^a^
RM	6.25 ^cd^	547 ^bc^	4.82 ^abc^	2.60 ^cde^	302 ^bc^	0.53 ^bc^
CT-b	6.43 ^cd^	602 ^de^	4.37 ^ab^	3.13 ^e^	301 ^bc^	0.48 ^b^
LLP	6.73 ^d^	660 ^g^	5.56 ^bcd^	2.76 ^de^	402 ^defg^	0.58 ^bcd^
Mean	5.88	578	5.61	2.42	352	0.58
SEM	0.080	10.4	0.151	0.059	11.00	0.013
Significance	0.001	<0.0001	<0.0001	<0.0001	<0.0001	<0.0001

AG, *Arachis glabrata*; AH, *Arachis hagenbackii*; CPb, *Centrosoma pubescene*; DB, *Dolichos biflorus*; CT-w, *Clitoria ternatea* (white); SSe, *Stylosanthas seabrana*; SH, *Stylosanthas hamata*; SSc, *Stylosanthas scabra*; SSco, *Stylosanthas scofield*; SV, *Stylosanthus viscosa*; MA, *Macroptilium atropurpureum*; AS, *Atylosia scarabaeoides*; DV, *Desmenths virgatus*; RM, *Rhynchosia minima*; CT-b; *Clitoria ternatea* (blue); LLP, *Lablab purpures*; ME, metabolisable energy; DMD, dry matter degradability; PF, partition factor (gas mL/mg DM); SFCA, short-chain fatty acids; MBM, microbial protein production; EMBP, efficient of microbial protein production; SEM, standard error of means; superscripted letters indicate significant differences in values.

**Table 7 animals-13-03676-t007:** Gas production, methane production and loss of energy as methane from in vitro fermentation of range of legumes in sheep inoculum.

Legumes	Gas mL/g	CH_4_ mL/g	Gas mL/g DDM	CH_4_ mL/g DDM	CH_4_% Gas	CH_4_%DE	CH_4_% ME
AG	110 ^cde^	15.2 ^d^	170 ^ab^	23.5 ^b^	13.9 ^bc^	5.77 ^ef^	7.03 ^ef^
AH	109 ^cde^	15.1 ^d^	164 ^ab^	22.7 ^b^	13.9 ^bc^	5.23 ^cdef^	6.37 ^cdef^
CP	99.1 ^bcd^	11.0 ^bc^	192 ^abc^	21.4 ^b^	11.1 ^abc^	4.14 ^abc^	5.04 ^abc^
DB	114 ^de^	13.0 ^cd^	199 ^abc^	22.6 ^b^	11.5 ^abc^	5.31 ^def^	6.47 ^def^
CT-w	81.3 ^b^	8.24 ^a^	145 ^a^	14.7 ^a^	10.1 ^ab^	3.23 ^a^	3.94 ^a^
SSe	108 ^cde^	11.5 ^bc^	188 ^abc^	20.1 ^ab^	10.8 ^abc^	6.10 ^f^	7.43 ^f^
SH	107 ^cde^	11.5 ^bc^	166 ^ab^	18.1 ^ab^	11.2 ^abc^	4.61 ^bcd^	5.61 ^bcd^
SSc	117 ^de^	12.8 ^cd^	186 ^abc^	20.4 ^b^	11.3 ^abc^	5.37 ^def^	6.54 ^def^
SSco	109 ^cde^	11.9 ^bc^	169 ^ab^	18.6 ^ab^	11.4 ^abc^	4.75 ^bcde^	5.79 ^bcde^
SV	115 ^de^	11.4 ^bc^	197 ^abc^	19.7 ^ab^	10.0 ^a^	4.60 ^bcd^	5.60 ^bcd^
MA	90.4 ^bc^	12.4 ^cd^	153 ^a^	21.0 ^b^	14.1 ^c^	4.72 ^bcde^	5.75 ^bcde^
AS	55.4 ^a^	9.1 ^ab^	188 ^abc^	23.2 ^b^	12.5 ^abc^	3.17 ^a^	3.86 ^a^
DV	122 ^def^	10.8 ^bc^	309 ^d^	26.4 ^c^	8.99 ^a^	3.24 ^a^	3.94 ^a^
RM	117 de	11.4 ^bc^	215 ^bc^	21.0 ^b^	9.72 ^a^	3.92 ^ab^	4.78 ^ab^
CT-b	141 ^f^	13.4 ^cd^	235 ^c^	22.3 ^b^	9.51 ^a^	5.27 ^cdef^	6.42 ^cdef^
LLP	124 ^ef^	12.8 ^cd^	190 ^abc^	19.4 ^ab^	10.4 ^abc^	4.49 ^bcd^	5.46 ^bcd^
Mean	107	12.0	192	21.0	11.3	4.62	5.63
SEM	1.80	0.219	5.86	0.495	0.315	0.134	0.163
Significance	<0.0001	<0.0001	<0.0001	0.010	0.041	<0.0001	<0.0001

AG, *Arachis glabrata*; AH, *Arachis hagenbackii*; CPb, *Centrosoma pubescene*; DB, *Dolichos biflorus*; CT-w, *Clitoria ternatea* (white); SSe, *Stylosanthas seabrana*; SH, *Stylosanthas hamata*; SSc, *Stylosanthas scabra*; SSco, *Stylosanthas scofield*; SV, *Stylosanthus viscosa*; MA, *Macroptilium atropurpureum*; AS, *Atylosia scarabaeoides*; DV, *Desmenths virgatus*; RM, *Rhynchosia minima*; CT-b; *Clitoria ternatea* (blue); LLP, *Lablab purpures*; DDM, dry matter; DE, digestible energy; ME, metabolisable energy; SEM, standard error of means; superscripted letters indicate significant differences in values.

## Data Availability

Data are contained within the article.

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
