# Peer review of "Nutrient and Rumen Fermentation Studies of Indian Pasture Legumes for Sustainable Animal Feed Utilisation in Semiarid Areas"

_animals, 2023, doi:10.3390/ani13233676_

Round 1
Reviewer 1 Report
Comments and Suggestions for Authors
The paper is the result of an extremely laborious laboratory activity that followed the nutritional evaluation of some important plant species for semi-arid areas. Also, the emission of gas in sheep following the consumption of these legumes was appreciated. The results demonstrate the correctness and experience of the authors in the choice of laboratory methods and the chosen experimental methodology. The paper makes numerous comparisons with the results obtained by other researchers in the field, which makes these results valuable for future scientific research.
- Originality/Novelty: The paper presents original results regarding legume pastures, which can be used in further research.
- Significance: All working assumptions have been specified and the results have been carefully analyzed. The research results are interpreted correctly. The conclusions are supported by the obtained results, but they should be presented in more detail, highlighting the contribution of the work to research in the field. Some graphs can be added to illustrate the results obtained based on the testing.
- Quality of presentation: The article is properly written, respecting the logical sequence of the sections. Data and analyzes are presented correctly, but some experimental data could be presented more suggestively in the form of graphs.
In the Discussions chapter, there is a quoted author who is not found in the References. Also, more explicit reference should be made to the results obtained by other researchers in the field (names, year, to highlight the topicality of the bibliographic references, but also the current number of the citation in the references can be mentioned).
The conclusions section should be enriched. Future research directions and what practical application scenarios you propose can also be included in the Discussions.
- Scientific soundness: The study conducted is well documented and will be useful for further research.
- Interest for readers: The obtained results will be of interest to the readers of the journal.
- General merit: The benefit of the publication of this work consists of a well-documented research on the use of plant species adapted to difficult climatic conditions, problems that are beginning to manifest in more and more regions of the world.
- English Level: The level of English language is advanced. Through the entire paper, the language was appropriate and understandable, being easy to follow the flow from the beginning.

Author Response
Dear Reviewer,
Thanks for your highly insightful comments and the time devoted to this manuscript. It
significantly helped me in the improvisation of the quality of our article. I have incorporated
all comments and advice in the main file of revised manuscript. In the following pages, I have
written the responses point by point for each comment.
Comment 1: All Latin names of plants must be written with italic characters (Page 2)
Response 1: Changed as per suggestion and incorporated into the final revised manuscript.
Comment 2: Berseen (section 2.7 in page 4).
Response 2: Changed to Berseem.
Comment 3: It is sufficient to indicate the citation number (section 4.2 page 10)
Response 3: Name of scientist deleted and only citation number kept, as per suggestion.
Comment 4: Buxton and Fales 1994 - not mentioned in References (section 4.2 page 10).
Response 4: Reference added.
Comment 5: You can put the name of the author of the work, in addition to the number
indicated in the references.
Response 5: author’s name inserted.
Reviewer 2 Report
Comments and Suggestions for Authors
First, I want to applaud the authors for looking at native forages in tropical and subtropical areas. This is an area that does need more study at both the botanical and animal level. However, that is where my admiration must stop. Many of my comments might be the result of the reviewer's poor understanding of the English used or the poor description of the procedures used, but in any case, they must both be corrected.
I have made a lot of suggestions for English improvement as well as pertinent scientific questions by highlighting in yellow and inserting comments on the enclosed document. Briefly, there needs to be a lot of work on getting agreement between subject and verb plurality as well as tense. I suggest working with someone who knows science and is a native English speaker. I suspect he or she could straighten this out in a week or two. What is less clear to me is how the experiment was designed. It sounds like the authors went to the plots and randomly took one sample. Yes they did the analysis in triplicate, but it is still only one sample. Therefore, we only have observations and not experiments. This does not mean that these observations do not have merit, but I do not believe they merit publication in a scientific journal. It would have been far better if they could have taken multiple samples over several times or in different seasons, but this does not appear to have happened. Because there is simply one observation per forage species, I cannot see how any statistics can be performed. There is not variation to evaluate.
Further, I cannot tell why the authors did the experiment seeing that they knew what the nutritional content of the forages were as they so tediously reference. Given the variance in tropical forages, it does not appear that they added any new knowledge to the subject. Given the amount of work which they did, this is a real shame.
Lastly, the abstract, even if I have misinterpreted the authors' procedures, must be laid out as follows: 1. a clear statement of study objectives (which they do provide); 2. a brief description of the study procedures (which) they do not provide); 3. a brief discussion of the relevant findings of the study (which they do not provide); 4. final conclusion from the study (which they kind of do). This is not the place for getting into the hard data. There is nothing new in this. It is part of every abstract I have written or read.

The English needs a moderate amount of work. My suggestions are included in the copies to the authors.
Author Response
Dear Reviewer,
Thanks for your highly insightful comments and the time devoted to this manuscript. It significantly helped me in the improvisation of the quality of our article. I have incorporated all comments and advice in the main file of the revised manuscript will be available to the editor. In the pdf version, I have written the responses point by point for each comment.
Thanks

Reviewer 3 Report
Comments and Suggestions for Authors
Age is undoubtedly one of the main factors that affects the quality and chemical composition of a forage, and the manuscript does not mention how old the forages were at the time of being analyzed (age of the forage after sowing in the case of annuals and age after previous cutting or sowing, if applicable, of perennials). If the age of the forages is different the response variables evaluated are not comparable and consequently the results of the study are meaningless. It is essential to clarify this point.
All tables must be self-explanatory, consequently the initials used to describe each species with its full name must appear at the bottom of tables 2,3,4, 5 and 6, as was done in table 1. In this same sense, there are several errors in the description of the variables evaluated, for example, in Table 2 the meaning of the initials CA, CB1, CB2 and CC is not explained at the bottom. Table 3 does not explain the meaning of the initials ME, DE, DMI, DDM and RFV and includes DM, EE, Lignin, NSC and OM which do not belong to this table. Furthermore, it is recommended to write them in the same order in which they appear in the table for better understanding.
It is more common in scientific writing to begin assigning the literals in descending order to the numerical value of the variable evaluated, that is, the highest value is assigned the literal a and so on, assigning the lowest value to the last literal used. In this manuscript the authors do it the other way around and, from my point of view it makes it a little more complicated to read the results from the tables. I would recommend reordering.
Given the large number of species and response variables evaluated, I recommend describing the results of each table in a more consistent way. Focus on pointing out those species with the lowest and highest values of each variable evaluated and in the discussion address whether the ranges found are within or outside the values reported by other studies.
The initial part of the conclusion is obvious: practically all legumes have a PC content greater than 10%. It should be concluded if there is one or several species or, as indicated in some parts of the document, a genus that stands out in all the variables evaluated and therefore its use is recommended in the region in which the study was carried out.
Author Response
Dear Reviewer,
Thanks for your highly insightful comments and the time devoted to this manuscript. It significantly helped me in the improvisation of the quality of our article. I have incorporated all comments and advice in the main file of the revised manuscript. In the following pages, I have written the responses point by point for each comment (attached).
Thanks

Reviewer 4 Report
Comments and Suggestions for Authors
The topic of the manuscript is pertinent and topical: the search for local plant protein for animal feed is an increasingly important issue.
However, the work has several limitations, particularly in terms of statistical analysis.
Several revisions are needed, particularly in the wording of the results and discussion.
You should make the changes requested in the appendix in order to significantly improve the manuscript.

Author Response

(The authors gave the same response as above.)

Reviewer 5 Report
Comments and Suggestions for Authors
The introduction is good, but it uses very old references. The aim of the work could be better defined.
In the first section of the material and methods, they present the results of the dry matter of the plants.
When determining DM, they say that the samples were first dried in the sun and then placed in an oven at 60 degrees. If the aim was to determine and carry out direct grazing, they shouldn't have wilted the plants. They don't explain when they weighed the samples to determine DM.
They mention that they used the leaves of the plants to produce gas, but they don't indicate the experimental design of the gas production.
Some of the results are inconsistent, for example, one of the plants shows metabolizable energy values higher than digestible energy.
The presentation of the results is very confusing. They report on practically all the results, and as they use 16 plants, it gets confusing at times. They should just highlight the most important results in each of the tables.
Some of the tables presented lack captions.
The discussion is all based on comparisons between the results obtained and those of other works, and they rarely justify the results obtained. They simply say that this value is higher than that found in article X, or lower than that found in article Y. They should justify why the values obtained are different in that region of India.
The conclusion is weak and does not answer the objective.
Author Response

(The authors gave the same response as above.)

Reviewer 6 Report
Comments and Suggestions for Authors
General Comments
This manuscript is potentially an outstanding summary of forage legumes on the Indian subcontinent. The paper contains detailed information on the chemical composition of various legume species analyzed for the typical components and in vitro dry matter (DM) digestibility among others.
However, despite the presence of a large amount of information, the paper has serious structural shortcomings that should be addressed before it should be considered for publication in any journal. The main issues are as follows:
1) The manuscript should contain line numbers to organize a review.
a. At the moment, a review is nearly impossible to perform.
2) The grammar and punctuation will need to be improved substantially.
a. There are multiple occasions where parentheses occur but without reason, articles (the, a) are missing and upper case and lower case are used in the wrong fashion.
3) Presentation of legume species analyzed
a. This needs to be done in a tabular fashion to provide the reader with an overview of the various annual and perennial species.
b. If you present 10+ species of any kind, I suggest not describing DM in one single sentence (last paragraph on page 2).
c. You will need to italicize the species names everywhere in the manuscript.
4) Methodologies
a. Legume sampling
i. There is little information provided on the origin of the samples.
1. At a minimum, planting time and general growth stage of the species should be provided as this will obviously affect chemical composition.
2. Since you present annual and perennial species, were these planted at the same time? What is the prevailing climate at your experiment station?
b. Chemical analyses
i. It is not clear form the paper what the value of methane measurements are with respect to digestibility and general use by animals.
c. Statistical analysis
i. At the minimum, the authors should present the plot layout the legume samples were obtained from, number of replications etc.
5) Graphs and Tables
a. There is no way the reader can follow the multitude of legumes species just from the abbreviations provided.
i. You should spell out the species in each table and make a distinction between annual and perennial species.
ii. Table captions have to extended with regard to standard information such as P-value significance, comparison in either rows or columns, acronyms used etc.
Comments on the Quality of English LanguageAs indicated in the comments above, the grammar and punctuation have to be improved substantially. This refers to upper/lower case spelling, missing articles, use of acronyms at first use, sentence structure, and use of parentheses. The quality of the language is high, but the editing before submission was not done in a way it should be.
Author Response

(The authors gave the same response as above.)

Round 2
Reviewer 2 Report
Comments and Suggestions for Authors
Please see the attached file for specific recommendations. This is 100% better than the first draft.

The English is still a huge problem for me. It needs a lot of work. I made some comments, but I did not have enough time to go over all of it. I do suggest you have a person who speaks English as a first language go over it.
Author Response
Dear Reviewer,
Thanks again for the inconsistent insightful comments and the time devoted to enhancing this manuscript's quality. Your suggestions and advice have been incorporated in the main file of the second time revised manuscript.
In the attached file I have written the responses point by point for each comment.
Thanks

Reviewer 3 Report
Comments and Suggestions for Authors
Indicate how many repetitions of each legume were used in the data analysis. Mention if the number differs between each of the analysis groups (chemistry, digestibility, minerals, etc.).
Author Response

(The authors gave the same response as above.)

Reviewer 5 Report
Comments and Suggestions for Authors
The authors have made most of the suggested corrections
Author Response
Dear Reviewer,
Thanks again for the inconsistent insightful comments and the time devoted to enhancing this manuscript's quality.
Thanks
Reviewer 6 Report
Comments and Suggestions for Authors
General Comments:
The paper improved substantially, but much of the added text needs some editing in terms of grammar.
L97: This is supposed to read “…..and Silvopasture…”
L206: No sure if 906 g/kg organic matter is correct
L206: I doubt CP was only 12.54 g/kg; it probably was 125.4 g/kg
L332: Again, I don’t think CP was just 14.7 g/kg, that would be 1.47%.
Comments on the Quality of English LanguageThe authors added several paragraphs to clarify their research and to add more detail where necessary. Just as the initial draft, many of those paragraphs will need to be edited again in terms of grammar and syntax.
Author Response

(The authors gave the same response as above.)
